# Physical Fundamentals of Biomaterials Surface Electrical Functionalization

**DOI:** 10.3390/ma13204575

**Published:** 2020-10-14

**Authors:** Karlis Baltacis, Vladimir Bystrov, Anna Bystrova, Yuri Dekhtyar, Talivaldis Freivalds, Jan Raines, Krista Rozenberga, Hermanis Sorokins, Martins Zeidaks

**Affiliations:** 1Riga Technical University, Kaļķu Street 1, LV-1568 Riga, Latvia; karlis.baltacis@1slimnica.lv (K.B.); jurijs.dehtjars@rtu.lv (Y.D.); jan4905@gmail.com (J.R.); krista.sidorova@gmail.com (K.R.); hermanis.sorokins@rtu.lv (H.S.); zdx@inbox.lv (M.Z.); 2Institute of Mathematical Problems of Biology—the Branch of Keldysh Institute of Applied Mathematics of Russian Academy of Sciences, Puschino, 142290 Moscow, Russia; bystrov@impb.ru; 3University of Latvia, Raiņa Bulvāris 19, LV-1586 Riga, Latvia; talivaldis.freivalds@lu.lv

**Keywords:** biomaterials, surface, electrical charge, functionalization, hydroxyapatite, roughness, point defects, oxygen vacancies

## Abstract

This article is focusing on electrical functionalization of biomaterial’s surface to enhance its biocompatibility. It is an overview of previously unpublished results from a series of experiments concerning the effects surface electrical functionalization can have on biological systems. *Saccharomyces cerevisiae* cells were used for biological experiments. The hydroxyapatite (HAp) specimens were used to investigate influence of structural point defects on the surface electrical charge. Threshold photoelectron emission spectroscopy was used to measure the electron work function of HAp and biologic samples. The density functional theory and its different approximations were used for the calculation of HAp structures with defects. It was shown that the electrical charge deposition on the semiconductor or dielectric substrate can be delivered because of production of the point defects in HAp structure. The spatial arrangements of various atoms of the HAp lattice, i.e., PO_4_ and OH groups, oxygen vacancies, interstitial H atoms, etc., give the instruments to deposit the electrical charge on the substrate. Immobilization of the microorganisms can be achieved on the even surface of the substrate, characterized with a couple of nanometer roughness. This cells attachment can be controlled because of the surface electrical functionalization (deposition of the electrical charge). A protein layer as a shield for the accumulated surface charge was considered, and it was shown that the protein layer having a thickness below 1 µm is not crucial to shield the electrical charge deposited on the substrate surface. Moreover, the influence of surface charge on the attachment of microorganisms, when the surface roughness is excluded, and the influence of controlled surface roughness on the attachment of microorganisms, when surface charge is constant, were also considered.

## 1. Introduction

Biocompatibility of biomaterials depends on their bulk and surface properties. The surfaces of these biomaterials are in direct contact with cells of a host organism. Since biocompatibility is paramount to the successful integration of biomaterials into the host organism, the biomaterial surface must be engineered in a way that suits the cells’ needs for proper attachment and subsequent proliferation.

Generally, this can be accomplished using two main approaches:the engineering of a surface’s geometry (morphology)the engineering of a surface’s physicochemical properties.

The first approach provides the surface with recesses and alcoves into which cells may become embedded in a way that is suitable for their survival, growth, and proliferation. As a result, new tissue is produced and the biomaterial becomes incorporated into the host organism. However, currently available studies have not suggested a clear correlation between the dimensions of the recesses and alcoves present on the surfaces of biomaterials (i.e., surface roughness) and the success of cellular attachment nor their further growth and proliferation [1].

The second approach emphasizes cell-surface attachment through electrostatic and van der Waals interactions, otherwise referred to as adhesion. For the first time, the fundamental physics of adhesion was described by the Nobel Prize winner (1962) Lev D. Landau and his coworker Boris V. Derjaguin [2]. The Landau–Derjaguin theory supposes that a particle can become adhered to a surface when the potential energy of the cell–surface interaction caused by the attractive Van der Waals and repulsive electrostatic energies reaches its minimum. Technologically, this can be accomplished by controlling the surface’s energy through the application of coatings or/and electrical charge deposition.

An overview of publication dynamics (Scopus search engine, key words: implant-surface-roughness-cell, implant-surface-coating-cell, and implant-surface-charge-cell) demonstrates (Figure 1) that the interest exhibited towards surface engineering techniques for cell–surface interaction studies, which use surface roughness processing, and the application of coatings was similar until the late 1990s.

Although the annual number of publications has been steadily increasing for both types of technologies, beyond the year 2000, interest in coating technologies outpaced the interest in roughness processing. As for surface charge engineering techniques, the annual number of publications in this field has been significantly lagging and is currently at the same level as roughness processing and coating technologies were 15–20 years ago. Nevertheless, the growth dynamics of the annual number of publications show an increase in interest in surface charge engineering similar to that of coating technologies and roughness processing 15–20 years ago (Figure 2).

This indicates that surface charge engineering has an optimistic future and could complement or even compete with conventional roughness processing and coating technologies for the improvement of cell–surface interactions. In anticipation of this, the present article discusses previously unpublished results of several experiments concerning biomaterial surface charge engineering with hydroxyapatite (HAp) being a prominent object of study due to its wide use as a biomaterial for bone substitution.

Presently, there are publications that demonstrate an influence of HAp electrical charge on cell attachment and proliferation [3,4,5,6], electrical charge engineering via electrical polarization [3], sample irradiation [4,7,8], modulation of surface roughness [6], doping [9], provision of single point defects (theoretical calculations) [10], and nanostructuring of the surface (Ti) [11]. There are also examples of the charge influence on wettability of even surfaces (PMMA, roughness 1–2 nm) [7] and HAp surface with microscale surface roughness [8].

However, there are still many questions that remain unresolved:there are no model experiments that decouple the influence of surface charge on attachment of microorganisms from surface roughness, i.e., cell deposition studies on smooth surfaces with evenly distributed surface charge are necessary.the influence of a protein layer (which forms around cells after they become attached to a surface within a host) on surface charge shielding.there are no model experiments that identify the influence of controlled surface roughness on the attachment of microorganisms when surface charge is constant, i.e., cell deposition studies on surfaces with controlled roughness are necessary.since real HAp ceramics have complex point defects in their structure, their influence on the accumulation of charge on the surfaces of HAp ceramics needs to be studied with numerical simulations preceding physical experiments.the influence of oxygen vacancies of HAp on its surface electrical charge remains unknown (HAp (Ca_5_(PO_4_)_3_OH) has the highest amount (60%) of oxygen atoms against the others); this should first be assessed using numerical simulations;experimental evidence that point defects of HAp influence its surface electrical charge.

The present article aims to answer these questions. To reach this, the article is structured in logically linked paragraphs.

First, the attachment of the microorganisms to an even charged surface is considered to get the direct experiment evidence that the electrical charge deposited on implanted material influences attachment of the microorganisms.

A surface of implanted material placed in a human organism is coated with proteins delivered with surrounding blood and lymph. In this case, a protein layer can shield the deposited electrical charge. This is considered at the nest step of the narrative.

The surface roughness is a conventional factor that is controlled to get the cell immobilized on the surface. The shapes of the roughness elements and their influence on immobilization are considered in the next paragraph.

Both theoretical and experimental results follow the above to demonstrate several possibilities of electrical charge induction because of the biomaterial controllable structural defects. Hydroxyapatite is used as the example.

## 2. Attachment of Microorganisms to the Even Charged Surface

### 2.1. Specimens and Methods

Cell immobilization substrates were prepared by cutting standard unfrosted microscope glass slides (Menzel Gläser, Thermo Scientific, Mwnzel, Germany) into pieces. Each 1 × 26 × 76 mm^3^ glass slide was cut by hand with a diamond scriber into (10 ± 1) × (13 ± 1) × (19 ± 1) mm^3^ individual samples further referred to as substrates. The surface roughness of the substrates was measured to be 1.23 ± 0.59 nm using atomic force microscopy in tapping mode (Solver–P-47 Pro, NT-MDT, Chernogolovko, Russia).

Before treatment with UV light, substrates were consequently washed in 96% ethanol (single washing, 1 min) and distillated water (double washing in separate vessels, 1 min each).

Surface charge deposition using substrate irradiation with ultraviolet (UV) light was performed as described in [4,7] using a UV spotlight source (LIGHTNINGCURE LC5 with a L8253 UV bulb, Hamamatsu, Japan). The distance between the source and the substrate surface was 30 cm. The exposure time was varied from 15 to 160 s (±1 s).

To identify the deposited charge, electron work function (φ) measurements on substrate surfaces were performed before and after irradiation [6]. For this, photoelectron emission spectra in the 3–6 eV photon energy range were measured under high-vacuum conditions. The value of φ is the minimal energy required for an electron to escape from a solid. Therefore, the surface electrical charge contributes to the value of φ, which increases when the charge is altered in the negative direction. The value of φ was identified as the photon energy when the emission current is equal to zero. The method and the spectrometer used are described in detail in [12].

Dried baker’s yeast “Rīgas Raugs” for experiments were bought from grocery store RIMI, located at Riga, Āzenes iela 5, Kurzemes rajons, Rīga, LV-1006 (Orkla ASA; SIA Lanordija, Ulbrokas, Riga, Latvia).

*Saccharomyces cerevisiae* cells were immobilized on the surfaces of the substrates after substrate irradiation. The time interval from irradiation to immobilization was around 3 h. After irradiation the substrates (3 substrates at one time) were placed in a 200 mL glass beaker after which 50 mL of yeast cell suspension (2.5 g of dried yeast cells per 50 mL of double-distilled water) were added to fully submerge the substrates. The flask was then shaken at room temperature for 60 min using an orbital shaker (OS-20, Biosan, Riga, Latvia). After that, the substrates were taken out from the beaker and washed in 250 mL of double-distilled water. During washing, the substrates were held by a pair of tweezers and shaken vigorously for 30 s. The washed substrates were dried in air at room temperature.

The number of cells attached to the substrates was estimated using optical microscopy (DM 1000, Leica, Wetzlar, Germany) equipped with a video system connected to a PC. The Image-Pro Plus Version 6.0. image processing and 2D analysis software was applied to analyze the images. The field of vision was the same for all measurements. The 30 analyzed fields of the 10 × 16 mm^2^ substrate area were chosen randomly (Excel RND function was employed) for each substrate. The optical magnification was 40×.

### 2.2. Results and Discussion

The achieved results (Figure 3) demonstrate that an increase in φ leads to an increase in the number of cells attached to the smooth surfaces of glass substrates.

Therefore, a negative surface electrical charge (increase of φ identifies that the surface charge becomes more negative) enhances attachment of *S. cerevisiae* yeast cells.

## 3. Protein Layer as a Surface Charge Shield

The electrical field induced by the presence of surface charge decreases with distance. In a human organism (blood electrolyte environment), the Debye layer has a thickness of around 0.01 nm [13]. This dimension is significantly less than that of a cell. Therefore, the cell–substrate surface charge interactions are facilitated only via the atomic or/and molecular residue structures located at the cell surface (membrane or wall).

Within 10 min after a biomaterial is inserted into a host organism, an organic coating (0.05–0.2 nm in thickness) composed of proteins that originate in the extracellular matrix (ECM) forms on the surface of the inserted material [14]. This means that the surface charge situated on the biomaterial surface is not directly available to the cells for interaction. On the other hand, a dielectric layer of adsorbed proteins has an amount of mobile charge carriers significantly smaller than that of blood. Therefore, the protein layer can be characterized with a Debye length that is significantly longer than the above. Thus, the protein layer can be polarized due to the presence of an electric field induced by surface charge.

The value of φ ≤ 0.1 eV can be induced in a HAp layer [15], that the photoelectron emits from has a thickness around 0.01–0.1 nm [16]. Because this thickness is significantly less compared with one of the protein films, the HAp layer can be considered as the thin plane. As the electrical permittivity of the protein is around 90 [17], the electrical field in a protein coating can be estimated as the electrical potential related to the thickness of the protein layer, i.e., 0.1 V/(0.05–0.2 nm)/90 = (0.5–2.2) × 10^−2^ V/nm. This field is several orders more intensive than the protein needs for polarization, i.e., 10^−6^–10^−5^ V/nm [18].

However, the protein layer polarization availability is not verified yet.

### 3.1. Specimens and Methods

Chicken egg white was deposited onto the glass substrate described in Section 2.1.

Before protein layer is deposited, the substrate was irradiated, using the UV radiation (Section 2.1) inducing the surface electrical charge. The UV exposure was equal to 900 ± 1 s. The charge induction was verified using the Kelvin probe spectroscopy technique (KPST). The atomic force microscope described in Section 2.1 was employed.

After that, the protein drop from a syringe was injected on the surface of the substrate. The latter was loaded on the handmade holder attached to the rotating disk of the centrifuge device Spin Coater VTC-200 (KISCO, New York, NY, USA). To get the protein layers having different thicknesses, the rotation of the disk and duration of procedure were varied in a 200–300 revolutions/min and 1–30 s, correspondingly. The deposited layers were dried in a room condition during 5 min. The thickness of the layers was measured using the atomic force microscope (contact mode) described in Section 2.1.

The electrical field induced on the deposited protein layer surface was identified using the above KPST.

### 3.2. Results and Discussion

The achieved results presented in Figure 4 demonstrates that the value of KPST falls down, when the protein thickness is >1 µm.

Following Figure 4, one can conclude that the substrate electrical charge can keep polarization of the protein deposited layer until its thickness is <l µm.

Therefore, the cell in the human organism directed to the biomaterial surface are expected to communicate with the electrically polarized surface of the deposited protein films that is induced because of the electrically charged surface of the substrate.

## 4. Controlled Surface Roughness to Attach Microorganisms at the Constant Surface Charge Condition

The experiment is aimed to explore the influence of the substrate surface roughness to attach cells. The study in intended to explore the influence of the valleys and peaks sizes of surface roughness to attach the cells.

### 4.1. Specimens and Methods

*Saccharomyces cerevisiae* was chosen for the experiments as the model. These cells are stronger than the human ones. Therefore, the cells are not deformed, when one observes using an optical microscopy, compared with the human ones. This is necessary to prevent the cell against capillary forces by the substrate roughness that are able to deform the microorganism and influence its attachment.

The substrates to adhere the cells were prepared as the set of the regularly situated parallelepiped pillars, each of them having a squared cross-section. The monocrystalline 470 μm thick silicon wafers (surface crystallographic orientation (100), electrical resistance 10 Ω) were used to fabricate the pillars. The wafers were washed in CARO solution and the SiO_2_ 0.6 μm thick layer was deposited (oxidation, oven CDOM-3, temperature *T* = 1130 °C, oxygen environment, exposure *t* = 2 h). The oxidation made the surface similar to the glassed one described in Section 2 and Section 3. After that, the photolithography was applied to reach the topology. The process was provided employing the Lada 1 (Ellar-Planar, Minsk, Belarus) equipment set instrumented with the wet-bench device for chemical etching of the photoresist. Then, plasma chemical etching was applied to reach the pillars. Because of this, the SiO_2_ layer was removed from the valleys, which was coated by the native ~1 nm thick silicon oxide grown during storage of the specimens in normal atmospheric conditions during a couple of weeks [19].

The processed wafers were split into the single substrates sized 5 × 5 mm^2^. For this, the mechanical diamond scriber was used.

The valleys between the pillars were expected to serve as the niches to locate the cells. Therefore, the pillars were heighted to 5.4–6.9 μm (measured using the scanning electron microscope (SEM) JEOL JSM-6400, Tokyo, Japan) as the cell (*Saccharomyces cerevisiae*) typically has the ellipsoidal shape, characterized with the large and small diameters 5–10 and 1–7 μm, correspondingly [20].

The distance between the pillars and their cross-section sizes was varied (Table 1).

The two types of the coupled experiments became available:equal cross-sections, however, different distances between the pillars (A + C).equal distances between the pillars, however, different cross-sections (A + B).

Figure 5 gives the images of the pillared structures.

The surfaces of the fabricated pillars were measured using the above atomic force microscope in tapping and amplitude modulated Kelvin probe force microscopy modes. The roughness on the pillars top was 1.1 ± 0.3 nm, and the KPST voltage had a value of 4.67 ± 0.11 V.

To identify properties of the valleys, they were measured at the outer area of the substrate edge; the roughness was equal to 4.0 ± 1.3 nm and the KPST voltage was 4.49 ± 0.06 V.

Both the roughness and the KPST voltage on the pillar tops and valleys were similar. The top roughness on the top was the same as that of glass substrates described in Section 2 and Section 3.

Before cell deposition, the substrates were washed in a series of solvents—agitated in acetone (ACS reagent, ≥99.5%, Sigma-Aldrich, Karriere, Germany) for 5 min, agitated in isopropyl alcohol (≥99.7%, FCC, FG, Sigma-Aldrich, Karriere, Germany) for 5 min, and then agitated in ethanol (96%, Sigma-Aldrich) for 5 min—after which, the substrates were rinsed three times in double-distilled water and dried in a semicovered Petri dish at room temperature until dry. After drying, the dish was fully covered and placed into storage.

The water suspension of the cell was prepared for deposition on the prepared substrates. A mass of 0.5 g of the cells was added to 150 mL of distilled water in a glass beaker and then mixed on a magnetic stirrer (MSH-300, Biosan, Riga, Latvia) until the suspension became uniformly visually opaque (~2 min). After that, a 1 mL sample of the suspension was taken to measure its optical density (at 600 nm with an UV–VIS spectrometer Spectronic Helios Gamma, Thermo Fisher, Waltham, MA, USA). The concentration of water/cell was adjusted to reach the absorbance of 0.187.

The substrates were located into the Petri dish containing the cell suspension. To keep the substrates in place and equidistant from the center of the dish, a dedicated sample holder was manufactured of an acrylic photopolymer using stereolithography (3D Printing UV Sensitive Resin (Basic), Photon, Anycubic, Shenzhen, China). The following printing parameters were used: layer thickness ~ 0.05 mm, normal exposure time ~ 10 s, off time ~ 10 s, bottom layer exposure time ~ 30 s, number of bottom layers ~ 6. After printing the sample holder was cured for 30 min both from the top and the bottom using a polychromatic UV source (Lightning Cure Spot UV source LC8 L9588, Hamamatsu, Japan). Distance from the UV source to the sample surface was kept at 30 cm. The holder fitted snuggly into a Petri dish having an internal diameter of 90 mm. The holder had 12 sockets for the single substrates located at 26.1 mm distance of the dish center to the center of the substrate.

The Petri dishes were filled in suspension to fully cover the substrate with liquid (~50 mL). Then, the dish was placed on an orbital shaker (OS-20, Biosan, Riga, Latvia) to be rotated at 50 RPM for 60 min. After that, the substrates were kept stationary for 20 min. Then, the liquid was slowly and carefully removed with a syringe and the samples were left to dry for 5 min. After that, 50 mL of distilled water was added into the Petri dish, which was then once again rotated at 50 RPM for 5 min. Lastly, the water was removed with a syringe and the samples were placed into a thermostat (58/350 LSN11, SNOL) for 15 min to dry at 30 ± 2 °C.

A number of the immobilized cells was analyzed using a Zeiss Jena NU-2 (Jena, Germany) optical microscope in reflected light mode at 125× magnification. The microscope was equipped with an ocular-mounted digital camera to acquire digital images. A custom-made motorized stage was fabricated to enable movement along the XY-plane with the uncertainty of ±10 μm. A single field of view (FOV) had an area of 417 × 238 mkm^2^. Imaging was performed in a sequential fashion for the whole of an immobilization substrate area, resulting in a 12 × 21 array of images from a single substrate, giving a total of 252 FOVs per substrate. When taking an image, the focus was kept in a plane at the bottom of the structures and adjusted, when it was necessary for getting detailed image of the cells.

The number of the attached cells was estimated as the ratio (R) of the area of the substrate occupied by the cells against the clean substrate. To identify the amounts of the cells attached separately to the pillar tops and valleys, a dedicated algorithm was developed using MathWorks MATLAB (version R2019b). A “mask” image of a clean substrate was subtracted from the image of having the deposited cells. The resulting image displayed only the cells attached to the surface of a substrate (Figure 6).

It was identified (Figure 6) that the cells are mainly attached at the edges (~650 μm width lane, ~10% of the substrate size) of the substrate C characterized with the narrowest valleys that are comparable with the dimensions of the cell. Therefore, to avoid the “edge effect” influence, the 650 μm lanes were excluded.

### 4.2. Results and Discussion

Figure 7 presents the example of the image of the substrates with the immobilized cells.

Following this result one can conclude that the cells prefer to be immunized in the valleys, that serve as the niches.

Figure 8 indicates the amounts of the attached cells on different substrates.

The results give the sense that the cells are trended to attach more effectively if the cross-section of the pillar is increased (cross-section of B is larger than A). This can be explained because the cells concentrated in the valleys serve as the pedestal to the other ones to make the bridges cover the pillars’ “roof” (Figure 9).

The cells are weakly attached if the valley width become narrow; the valley of the C substrate is narrower than of the A one (Figure 8).

## 5. Complex Point Defects of HAp toward Induction of Its Surface Electrical Charge

### 5.1. The Main Particularities of HAp Structure

HAp has a complex structure:as a rule, either ceramics or a powder of various kinds of micro- and nanoparticles with a set of grains of different sizes is used (with its own statistical spread around some average size).basically, in this case, the crystal structure of HAp grains has a hexagonal phase (P6_3_), stochastically disordered with respect to the orientation of OH groups (dipoles) in OH-channels, although they can be ordered (that is, parallel oriented OH structures), as well as monoclinic phases (P2_1_).the composition of hydroxyapatite is usually not stoichiometric (the ratio Ca/P = 1.67 [21,22] exists only in an ideal stoichiometric lattice).the main thing is that in this case, HAp has many different structural defects and this structural imperfection and inhomogeneity of it leads both to a scatter of its physical properties observed in the experiments and to noticeable differences in its physical properties from those theoretically calculated data by various methods for periodic crystal structures of HAp without defects.

It turns out that the electrical properties of HAp are primarily determined by the totality of its defects, and they create, e.g., the variability of the band gap Eg and optical absorption, as well as the work function of the electron φ, which do not coincide at all in these parameters with data for the usual ideal HAp lattice. Here arises the influence of such complex point defects of HAp on its surface electrical charge as the real HAp has complex point defects.

The influence of oxygen vacancies of HAp (Ca_5_(PO_4_)_3_OH) on its surface electrical charge can be the highest, as the amount of oxygen atoms is the largest against the other ones [23]. Complex OH-vacancies play an important role too, as well as H-interstitials and H-filling of unsaturated hydrogen bonds.

HAp can be considered as a wide-gap semiconductor [24,25]. Using methods of surface photovoltage spectroscopy (SPS), it was established that HAp ceramics samples (investigated in the research [24,25]) is a p-type wide band gap semiconductor with originally positive surface band-bending and several electron/hole bulk and surface states. In this case, the defects of its structure will change both the width of the forbidden gap (*E*g) itself and create additional energy levels Ei within this *E*g = *E*c − *E*v (where *E*c is the bottom of conduction band and *E*v is the top of valence band). These levels can be filled with electrons and holes, depending on the position of the Fermi level, and thus determine the type of conductivity of HAp—hole (p-type) or electronic (n-type) [26]. This, in turn, plays an important role in the distribution of the electric potential, including over the HAp surface. In addition, the position of the levels *E*i, as well as the top of the valence band *E*v, affects the electron work function φ = *E*_0_ − *E*v = *E*g + χ (where *E*_0_—vacuum level and χ—electron affinity).

HAp structure, used for studying the influence of defects on its properties, are primarily based on its initially pure structural phase-hexagonal, with a unit cell, consisting of 44 atoms and containing structural OH-channels from 2 hydroxyl OH groups in each unit [22,23]. Depending on the orientation of these OH groups, the cells can have different symmetry groups (Table 2): P6_3_/m—for a disordered phase (when the orientation of OH groups is random) and P6_3_—for an ordered phase (when the orientation of OH groups is parallel and directed in one direction, which creates its own internal polarization similar to ferroelectrics (see below, e.g., in [27]).

The same applies to the monoclinic phase of HAp—there may be a disordered phase P2_1_/b (when the orientation of OH groups in neighboring OH channels is opposite and the total polarization is zero) and an ordered phase P2_1_ (when OH groups are directed in parallel and in one direction, which creates a net of the total polarization). But, here, we consider mainly hexagonal phase. Figure 10 shows (a) main hexagonal unit cell structure and (b) several modelled defects in this HAp unit cell, which influence on HAp properties changes that are studied and discussed below.

There are several widely used approaches to calculate crystalline structure and its defects. The main one uses various approximations of the density functional theory (DFT) and was developed in the works of Kohn and Sham (KS) [29,30]. This theoretical approach has different approximations. One of them is the local density approximation (LDA) [31,32]. This is implemented in the AIMPRO software package [32], in which calculations of HAp (initial pure ideal and with defects) were performed [22,28]. Another one is the generalized gradient approximated (GGA) approach, originally developed by Perdew, Burke, and Ernzerhof (because named “PBE”) [33], which turns out to be more general and accurate in many cases and that was also used in HAp calculations [27,34,35] using the VASP software package [36,37,38].

In addition to GGA (PBE) the B3LYP (Becke, 3-parameter, Lee–Yang–Parr hybrid functional) was used [39,40,41] to get more accurate account of electronic exchange and correlation potentials in hybrid DFT functionals. Studies show that this is more consistent with HAp, since its structure has a molecular-like structure: isolated PO_4_ tetrahedra, calcium columns and a hydroxyl channel. The structure of the electronic bands is rather flat, which confirms the “molecular” nature of HAp. This is because the B3LYP functional was developed to describe molecules. This exchange correlation functional allows also to describe the properties of electronic excitation and the energy of defect formation. Results of calculating the structure and properties of HAp by other authors are performed in [42,43,44,45,46,47,48,49,50].

The questions now are how defects change this entire band energy structure of the states of electrons and how close it will be to the experimental values, when creating certain defects in HAp structure. For calculations at the first stage [22,28], the defects simulated were created by vacancies and interstices into one hexagonal unit cell of HAp consisting of 44 atoms. Figure 10b schematically shows examples of simulated HAp structures with defects such as oxygen vacancies from the OH and PO_4_ groups, as well as a vacancy of the entire OH group.

As a result of LDA calculations (using AIMPRO software [32], version aimpro-2.3.13b2 and/or aimpro-2.3.13b -par), the distributions of the density of states of electrons by energy (DOS) corresponding to the filling of energy levels in the scheme with electrons and the energies of HAp according to the band theory [26] were obtained [22,28]. This makes it possible to determine the main parameters: the position of the top of the valence band Ev, the bottom of the conduction band Ec, and the value of the band gap Eg = Ec − Ev, as well as the position of additional energy levels Ei induced by defects in the internal range of Eg.

Figure 11a,b shows examples of DOS for pure defect-free HAp (taking into account both deeper energy levels in the valence band and in the vicinity of the forbidden band) obtained in LDA calculations. Similarly, GGA calculations were carried out using VASP software [38] (vasp.5.4.1.—vasp.5.4.4. versions). Figure 11c shows results of DOS from GGA (with PBE functional). In Figure 11e,f, examples of DOS in the results of GGA (PBE) calculations are given for defects such as O vacancies from the OH group, O vacancies from the PO_4_ group, and a full vacancy of the OH group (Table 3).

Note that the calculations using the LDA and GGA (PBE) approximations generally do not fundamentally differ here, the difference is only in the energies—GGA (PBE) gives a larger value of the band gap Eg ~ 5.26 ± 0.05 eV (for pure HAp) compared to calculations by the LDA method, which give Eg ~ 4.6 ± 0.05 eV. These data are close to results of other authors [42,43,44,45]. The results obtained also lead to a small shift in all level energies for defects in the calculations of the GGA (PBE) methods relative to LDA (Table 3). Both methods predict the changes in band gap Eg and of the electron work function, which can be experimentally measured.

Further development of the calculations was the transition of calculations from one unit cell to a super-cell consisting from of 2 × 2 × 2 = 8 unit cells, (space group P6_3_, comprising a total of 352 atoms), as well as the application of the hybrid functional PBE in combination with B3LYP using DFT method [34,35]. This made a possibility to classify different types of oxygen vacancies of the PO_4_ group more correctly and accurately, which increases the calculation accuracy, and to highlight new more complex types of defects—extended charged complex oxygen vacancies.

### 5.2. Oxygen Originated Complex HAp Defects

Performing these series of calculations, it was found that different types of defects arise if there is symmetry of the atomic group associated with different spatial arrangement of various atoms in the PO_4_ group. Besides, usual oxygen vacancy V_O_ arises complex extended defects that depend on their charge state Q = 0, +1, +2. (Figure 12) [27,35]. The O vacancy of OH group influences the formation of new complex defects because of charge Q variation.

Crystalline HAp has several oxygen atoms (from the PO_4_ group and the OH group) in four symmetries at the equivalent sites (they are marked as I-IV). Atoms I-III are located in the phosphate units PO_4_, and oxygen atoms of type IV are in the hydroxyl group anions. They are named “O(I)–O(IV)” (“Bulk HAp” in Figure 12, where only corresponding vacancies V_O(I)_ and V_O(IV)_ are shown). It should be noted that PO_4_ groups have two almost symmetric types of oxygen atoms O(III). This is presented as schematic overlap of oxygen atoms on the upper row in the “Bulk HAp” part of Figure 12.

We consider mainly vacancy structures with lower energy, shown in Figure 12a–e. The formation of pyramidal PO_3_ structure is schematically shown in Figure 12a. This structure is obtained from PO_4_ after exclusion of oxygen O(I) atom (bright red-colored circle on the “Bulk HAp” part of Figure 12). The resulted vacancy is presented in Figure 12a (“Defective HAp”).

The similar structures were obtained for V_O(II)_ and V_O(III)_ vacancies. An isosurface of the electron density corresponding to the highest occupied electron state for the neutral charge state is presented in Figure 13a. There, the case of V_O(III)_^0^ is considered as an example.

Figure 12b represents the case without an oxygen O(IV) atom, which leads to an insulated H atom in the OH channel of HAp. Figure 13b represents the electron density isosurfaces corresponding to the highest occupied electron state of V_O(IV)_^0^. The formation of a hydride anion in the OH channel is clearly shown here. The full OH-vacancy defect is shown in Figure 12e [34].

In the cases of the V_O(I)_–V_O(IV)_ defects, all atoms stay nearby to their original crystalline coordinates. That is why, the subscripted O(I)–O(IV) labels are used for identification of their structures. The extended structures were also found for the O vacancy in HAp structure. There are two types of such vacancies. The first one is a pair of neighboring oxygen vacancies connected by an O-interstitial, 2V_O_ + O. These defects are denoted as V_O(A)_ and V_O(C)_ (Figure 12c,e).

The second type is a complex consisting of an OH–vacancy and an H–interstitial, V_OH_ + H. This defect is denoted as V_O(B)_ (Figure 12d). The highest occupied electron state of the extended structures is shown in Figure 13c,d. They overlap the void regions of the HAp crystal structure or the vacant volume of the HAp OH-channel. Therefore, they can be considered as the donors or moiety with strong resonance interacting with conduction band states [34].

The charge state of vacancy V_O_ defects strongly influence their structure and stability. The oxygen V_O(I)_−V_O(IV)_ defects are stable only in case of neutral and single positive vacancies. Conversely, structures V_O(I)_−V_O(IV)_ are unstable. They modify to an extended configuration in cases of charge state with *q* = +2.

The resulted severe relaxation can be explained by the electron transfer from a neighboring PO_4_^3−^ anion to the empty P(sp^3^) orbital of the (PO_3_^−^)_PO4_^2+^ moiety in the V_O(I)_^2+^ or in the V_O(III)_^2+^ (Figure 13a) with the subsequent formation of a new chemical P−O bond. Figure 12c,e shows the final extended complex configuration of (PO_3_^2−^−O−PO_3_^2−^)_2(PO4)_^2+^ structure.

#### 5.2.1. Kohn−Sham Energy Levels of Neutral V_O_ Defects

Inspection of the Kohn−Sham eigenvalues at *k* = Γ confirmed that neutral vacancies V_O_ defects are all donors [27,35]. The calculations performed in these cases, considering the even more accurate B3LYP calculation scheme, showed similar shifts in the energy levels *E*i produced by oxygen vacancies of the PO_4_ and OH groups. The main contribution of these defects, which determines the change in optical properties (from *E*g of initial HAp without defects on the E_ig_^0^ = Ec* − Ei for optical excitation of electrons from energy level Ei to the conductive band Ec* and electron trapping Ei − Ev* from valence band Ev* to Ei) and the change in the work function Δφ ~ ΔEg = Eg* − Eg, remains at the similar level of ~ 0.1–1 eV (Table 4), despite the differences in the calculations of the band gap Eg of initial HAp structure without defects by various approaches PBE and B3LYP.

As it can be seen, regardless of the method of calculation, oxygen vacancies of the OH and PO_4_ groups (in the absence of charge in these defects, i.e., at Q = 0) form a group of energy levels (Figure 14) located close to the top of the valence band and they are donors electrons [35] (electron acceptors, i.e., levels with a negative charge, were not found).

In this case, the shift of the levels of oxygen vacancies in the PO_4_ group is ~1.15–1.65 eV upward from Ev and for a vacancy from the OH group, it is ~0.4–0.7 eV, and this also corresponds to a change in the electron work function during the formation of such defects. As everyone can see, the positions of these energy levels are not very different (especially in the case of PBE calculations), although in the case of B3LYP, these levels are slightly higher than Ev in the direct case. In any case, these deviations are within 1 eV.

It is also important to note that such levels that are close at the top of the valence band have recently been observed in experiments on photoelectron emission spectroscopy, and the work function of a photoelectron from HAp was measured for various external influences [51,52]. In addition, in [52], it was noted that such energy levels (and photoelectrons emitted from these energy levels) can arise under a number of actions on HAp samples (heating and annealing, gamma irradiation, microwave effects, and combined hydrogenation with microwave radiation). In these cases, a sufficiently large number of oxygen vacancies (also in the OH group) can be induced having the lowest energies levels measured from the top of the valence band (Table 3 and Figure 14).

It was also shown in [35] that charged oxygen vacancies also form more complex defects—extended or bridging (in the case of a charge Q = +2, Figure 12), the latter reconstructing to point-like defects (at Q = +1 and = 0). This rearrangement from extended to point defects occurs with bond breaking and causes optical absorption effects. This transition leads to spontaneous rupture of bridging P–O–P or O–H–O bonds at extended defects. In addition, this transition, most likely, explains the onset of absorption at 3.4–4.0 eV for observing photocatalysis under constant ultraviolet illumination [28]. It is important that the stability of defects and these structure types strongly depend on the charge states.

#### 5.2.2. Generation of Defects and Dependence on Their Charge State

Depending on the total charge level and on the concentration of charges in the HAp and its conductivity (p- or n-type), the formation of such defects occurs in different ways (see Figure 15 and Figure 16) [34,35].

Depending on the position of the Fermi level, which corresponds to the conductivity type, a structure of oxygen vacancies is changed. It is important that single positively charged defects are metastable. Considering the structures with the lowest energy for each state of charge, one can find that the level of the transition from *Q* = 0 to *Q* = + 2 (0/2 +) is at the intersection point (Figure 15). The location of the energy level at the intersection point turns out to be at E (0/2+) − Ev = 2.65 eV [35].

Therefore, the oxygen vacancies in case of p-type HAp are likely to assume double-positive extended (bridging) structures V_O(B)_^2+^ or V_O(C)_^2+^. This defect will be presented in form of an insulated H-hydride in the OH-channel (i.e., V_O(IV)_^0^) [35], in case of the n-type HAp and in the intermediate type.

At the same time, the defects such as vacancies of the entire OH group can be formed and exist in different intervals, depending on the adopted approximation in the calculations (Figure 16 [34]). However, the charged state V_OH_^+^ can exist in a rather wide range of Fermi energies from 0 to practically ~ 4 eV [34] (measured from the top of the valence band), i.e., in the p-type HAp (Figure 16). This is consistent with the observations of OH vacancies in HAp samples in [25,51,52].

## 6. Engineering of Surface Electrical Charge of HAp due to Its Structural Imperfections

### 6.1. Specimens and Methods

The pressed ceramic nanopowder HAp specimens used for the experiments had cylinder-shaped tablets with a diameter of 5 mm and thickness of 2 mm. The specimens were supplied from the collection of the PERCERAMICS project [53] that employed the technology described in [25,53,54]. Formation of HAp canalled ceramics specimens was done using the sol–gel technology. The samples were sintered under the temperature of about 1200 °C in the furnace in air media. The samples had the porosity of 10–40%. The data obtained using the scanning electron microscope T200–JEOL (Tokyo, Japan) demonstrated that the ceramics assembled of ∼20 nm sized nanoparticles (NP) consisted of NP aggregates clustered in ∼100–300 nm [25,53,54]. The channels in the ceramics had the diameter of 0.2–6.0 μm and the length of 25–85 μm [53]. The chemical content and structure of the samples corresponded to hexagonal Hap, identified using X-ray diffraction (diffractometer DRON-3, Burevestnik, St-Petersburg, Russia) [53,54]. The data were in accordance with the database of the American Mineralogical Society [23].

Charge deposition in HAp is possible by the production of OH−, O−, or H− vacancies or by hydrogen filling in unsaturated hydrogen bonds (Section 5).

To generate the structural defects, different treatments [25,55,56] were employed (Table 5).

The following parameters were chosen for applied treatments in accordance with [25,55,56]:Annealing:vacuum condition: 0.001–0.00115 Pa.temperature: 542–546 °C.heating rate: 15 °C/min.annealing time: 30 min.cooling down to room temperature.
Gamma-ray irradiation:photon energy: 18 MeV.the dose rate: 1000 MU/min.the ionizing radiation dose: 10 Gy.distance from the radiation source to sample: 1 m.radiation field: 10 × 10 cm^2^.
Hydrogenation:temperature: 23 °C.pressure of hydrogen: 60 ± 2 atmospheres.hydrogenation time: 6 h.
Microwave irradiation:power: 800 W.working time: 6.5 min.


The following equipment was employed to process HAp:Annealing was provided in a vacuum condition (10^−4^ Pa) using the custom-made equipment [12].Gamma-ray irradiation: linear accelerator CLINAC, Varian Medical Systems Inc., Charlottesville, VA, USA.Hydrogenation: autoclave, HPM-P, PREMEX, Lyss, Switzerland.Microwave (MW) irradiation: microwave oven, SAMSUNG, Seoul, South Korea.

The threshold photoelectron emission spectroscopy (PE) was used to measure φ of HAp. It is generally known that φ is identified as the photon energy *h*ν, when the electron emission current (I) is equal to zero (Equation (1)):I ~ (*h*ν−φ)^m^,(1)
where m is the power index.

The I(*h*ν) spectrum can be contributed by electrons emitted alongside from different edges, i.e., local level, top of the valance band, etc. In this case, the I(*h*ν) diagram is at different ranges of *h*ν and is characterized with the specific m indexes, each of them linking emission from the particular edge. To identify their energies, the I(*h*ν) spectrum starting from the smallest value of φ (φ0) was deconvoluted, and φ were identified for each part of the I(*h*ν) diagram. The enquired partial values of φ were indexed as φ0, φ1, etc.

The plan of the experiment is presented in Table 6.

### 6.2. Results and Discussion

The presence of defects in the samples was identified referencing to the calculated φ (Table 3). The group 1 (Table 6) of the samples was separated into the subgroups, all samples of the subgroup having the specific of φ (Table 3) characterizing the specific defects. The subgrouping is provided in Table 7. The experimental results here and below are in accordance with the calculated data (Section 5), the uncertainty being around ± 2.2%.

After annealing of the subgroups (Table 7), the samples of each subgroup were again collected into the sub-subgroups (Table 8) according to the calculated φ values (Table 3).

Following the data of Table 8, one can conclude that:the hydrogen atoms are knocked out during annealing and then captured by unsaturated hydrogen bonds that were natively present in the unit cell (sub-subgroups 1.1, 2.2, 5.1, and 6.1).the oxygen atoms are knocked out during annealing (sub-subgroups 1.2, 2.1, 3.2, and 6.1).the hydrogen atoms are knocked out during annealing without following capturing by unsaturated hydrogen bonds that were natively present in the unit cell (sub-subgroup 1.1).the OH-groups are knocked out during annealing (sub-subgroups 1.2, 2.1, and 3.1).

Analogously with the samples of group 1, the group 2 (Table 6) of the samples was separated into the subgroups after annealing. The subgrouping is provided in Table 9. Before annealing, the measured φ of samples of both subgroups were correspond to calculated value of HAp samples with OH–vacancies.

After gamma-ray irradiation, the measured φ value of samples of the 2.1 subgroup (Table 9) was 5.00 ± 0.04 eV, which corresponds to the calculated value of φ = 4.92 ± 0.11 eV of HAp with H–vacancy, and the measured φ value of samples of the 2.2 subgroup was 5.40 ± 0.04 eV, which corresponds to the calculated value of φ = 5.49 ± 0.21 eV of HAp with OH-vacancy. It means that gamma-ray irradiation not only produces H–vacancies, how it was predicted, but also produces OH-vacancies in the structure of the investigated samples by releasing OH–groups from the HAp OH-channel structure. After gamma-ray irradiation the expected (Table 5 [55]) presence of O-vacancies was not detected, probably because the chosen parameters of irradiation were enough to release full OH-groups, but not enough to release and separate oxygen atoms from OH-groups.

Analogously with the samples of group 1, the group 3 (Table 6) of the samples was separated into the subgroups after annealing. The subgrouping is provided in Table 10. Before annealing, the measured EWF of samples of both subgroups were correspond to calculated value of HAp sample with O-vacancy or H-interstitial and H-vacancy.

After hydrogenation, the measured φ value of samples of the 3.1 subgroup was 5.20 ± 0.04 eV, which corresponds to the calculated value of 5.12 ± 0.11 eV (Table 3) of HAp with H-interstitial, and the measured φ value of samples of the 3.2 subgroup was 5.39 ± 0.08 eV, which corresponds to the calculated φ value of HAp with OH–vacancy (5.49 ± 0.21 eV). It means that hydrogenation under chosen parameters not only produces H-interstitials but also fulfilled H-vacancies in the structure of the investigated HAp samples.

Analogously with the samples of group 1, the group 4 (Table 6) of the samples was separated into the subgroups. The subgrouping is provided in Table 11.

After annealing, φ0 = 5.12 eV and φ1 = 5.33 eV from the samples of the 4.1 subgroup were registered, where φ0 corresponds to the calculated φ value of HAp with O–vacancy and φ1 corresponds to the calculated φ value of HAp with OH–vacancy.

After annealing φ0 = 5.01 eV and φ1 = 5.13 eV from the samples of the 4.2 subgroup were registered, where φ0 corresponds to the calculated φ value of HAp with H–vacancy and φ1 corresponds to the calculated φ value of HAp with O–vacancy.

After microwave irradiation, the measured φ values of the samples of the 4.1 subgroup (5.26 ± 0.04 eV) began to correspond to the calculated φ values of HAp with O–vacancy (5.15 ± 0.11 eV) or H–interstitial (5.12 ± 0.11 eV) and the measured φ values of the samples of the 4.2 subgroup (5.30 ± 0.04 eV) began to correspond to the calculated φ value of HAp with OH–vacancy (5.49 ± 0.21 eV). It means that microwave irradiation not only influence formation of H–interstitials but also produces OH–vacancies and O–vacancies in the structure of the investigated HAp samples.

The group 5 (Table 6) of the samples was separated into the subgroups analogously to the sample group 1. The subgrouping is provided in Table 12.

After annealing, φ0 = 5.07 eV and φ1 = 5.31 eV from the samples of the 5.1 subgroup were registered, where φ0 corresponds to the calculated φ value of HAp with O–vacancy and φ1 corresponds to the calculated φ value of HAp with OH–vacancy.

After annealing, the measured φ values of the samples of the 5.2 subgroup began to correspond to the calculated φ value of HAp with O–vacancy.

After hydrogenation and microwave irradiation, φ0 = 5.00 eV and φ1 = 5.16 eV from the samples of the 5.1 subgroup were registered, where φ0 corresponds to the calculated φ value of HAp with H–vacancy and φ1 corresponds to the calculated φ value of HAp with O–vacancy or H-interstitials. After hydrogenation and microwave irradiation, the measured φ values of the samples of the 5.2 subgroup began to correspond to the calculated φ values of HAp with O–vacancy or H–interstitial. It means that H–interstitials or O–vacancies and H–vacancies produce inside HAp samples as the result of applied hydrogenation and following microwave irradiation.

## 7. Conclusions

Immobilization of the microorganisms can be achieved on the even surface of the substrate, characterized with a couple of nanometer roughness. The attachment of the cells can be controlled because of the surface electrical functionalization (deposition of the electrical charge).The protein layer having a thickness below 1 µm is not crucial to shield the electrical charge deposited on the substrate surface.The micrometric roughness of the substrate surface attaches the cells, the capillary phenomenon can play an important role to get the cell into the roughness valleys. The cells located at the valleys connect them with the bridges that stretch over the morphology pinnacles and cover the latter.The electrical charge deposition on the semiconductor or dielectric substrate can be delivered because of production of the point defects, hydroxyapatite being the example.The spatial arrangements of various atoms of the hydroxyapatite lattice, i.e., PO_4_ and OH groups, oxygen vacancies, interstitial H atoms, etc., give the instruments to deposit the electrical charge on the substrate.

## Figures and Tables

**Figure 1 materials-13-04575-f001:**
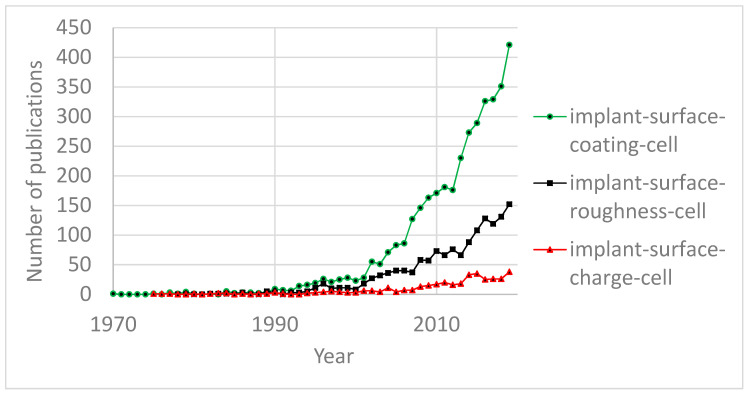
Annual number of publications.

**Figure 2 materials-13-04575-f002:**
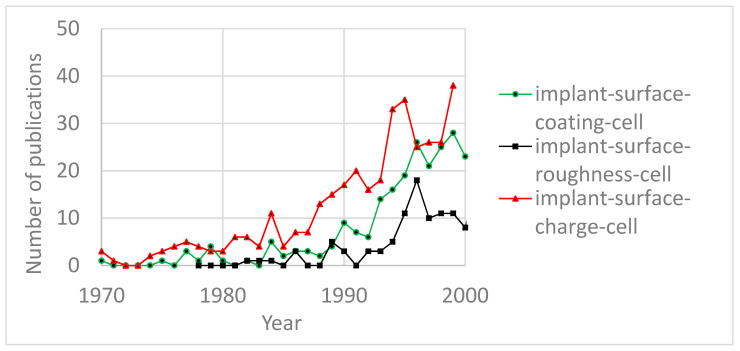
Annual number of publications: the “implant-surface-charge-cell” diagram is shifted back for 20 years.

**Figure 3 materials-13-04575-f003:**
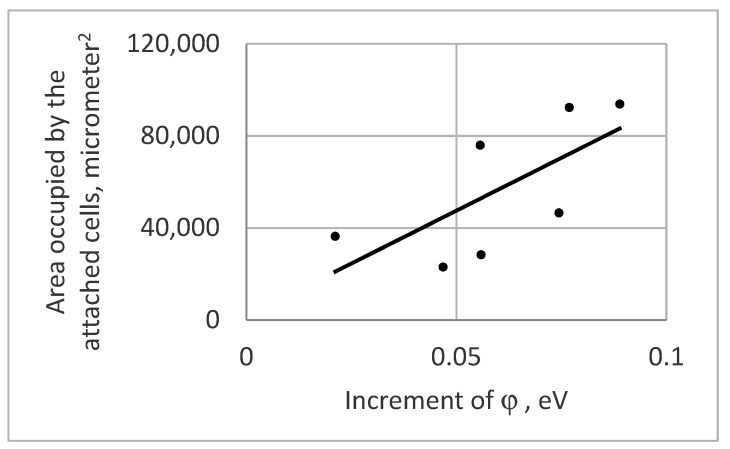
The trend of the amount of the attached cells in dependence on the increase of φ.

**Figure 4 materials-13-04575-f004:**
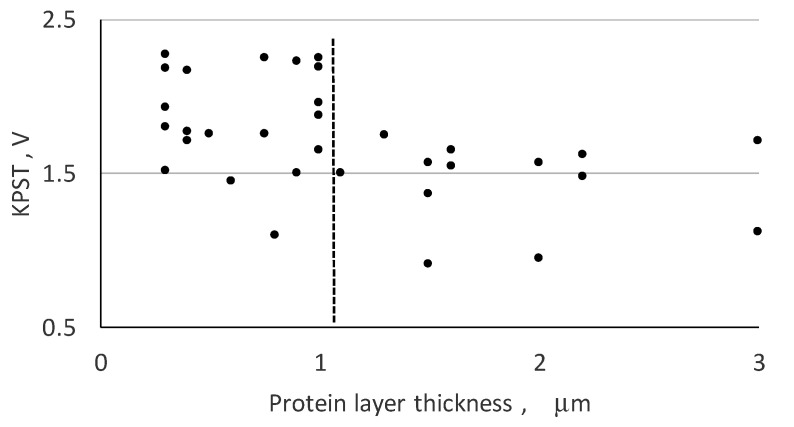
KPST voltage influenced by the thickness of the protein.

**Figure 5 materials-13-04575-f005:**
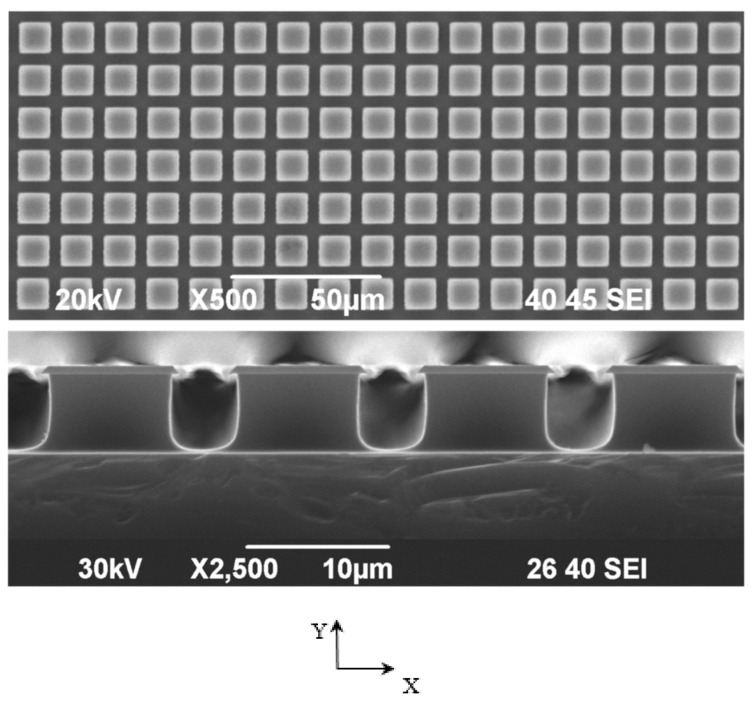
SEM images of the pillared structures (C substrate as the example).

**Figure 6 materials-13-04575-f006:**
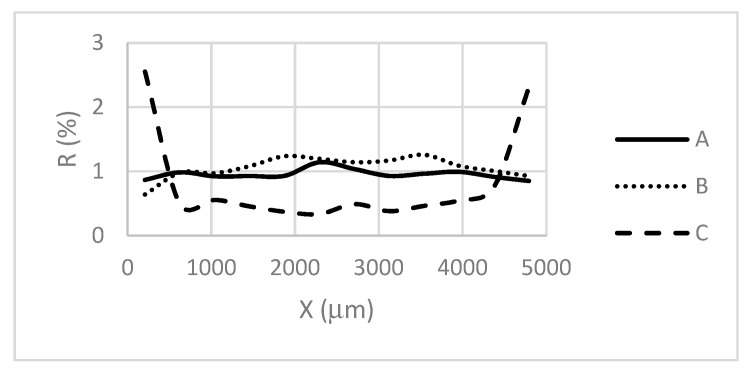
Deposition of the cells along X axis of the substrates.

**Figure 7 materials-13-04575-f007:**
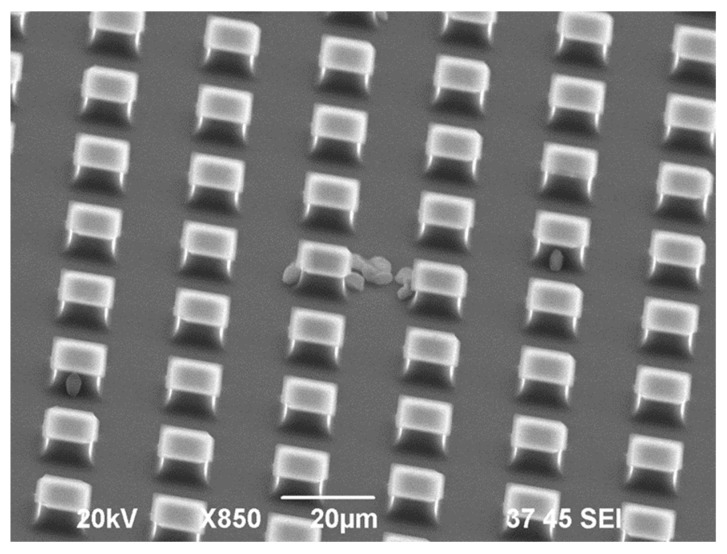
SEM image of the substrate with the immobilized cells.

**Figure 8 materials-13-04575-f008:**
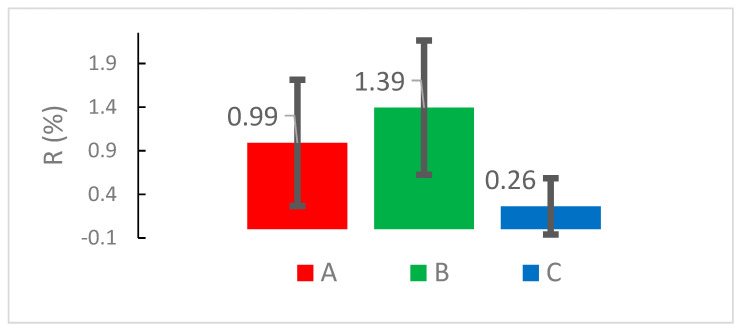
The amounts of the attached cells on different substrates.

**Figure 9 materials-13-04575-f009:**
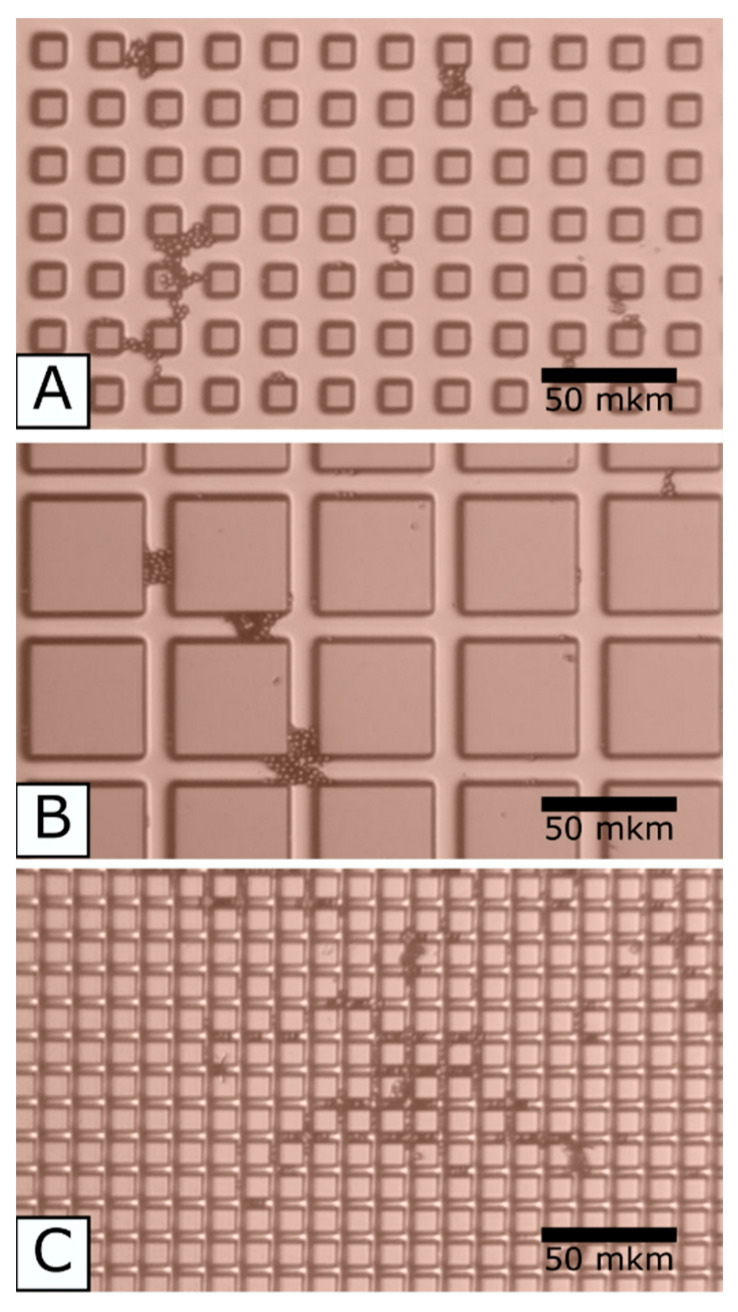
Optical microscopy images of the cells attached to the different substrates.

**Figure 10 materials-13-04575-f010:**
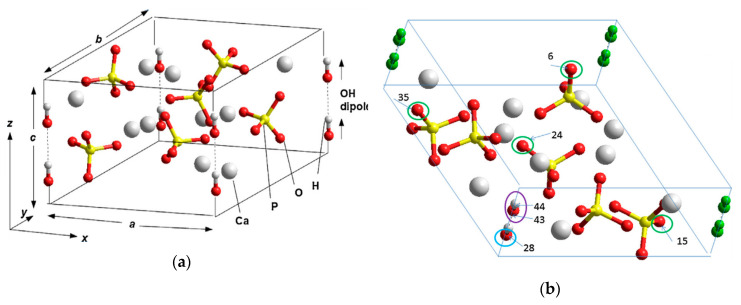
(**a**) HAp unit cell of hexagonal P6_3_ [22,23]. (**b**) Schematic of the atoms selected for modelling of various vacancy defects in the HAp hexagonal unit cell with 44 atoms [28]. Indicated are: (1) in blue—oxygen atom number 28 from an OH group for creation of an O vacancy in the OH-channel; (2) in purple—OH group with atom numbers 43 and 44 for creation of a complete OH vacancy; (3) in green—oxygen atoms with numbers 6, 15, 24, and 35 from various differently positioned PO_4_ groups corresponding to an O vacancy in PO_4_ groups [22,28].

**Figure 11 materials-13-04575-f011:**
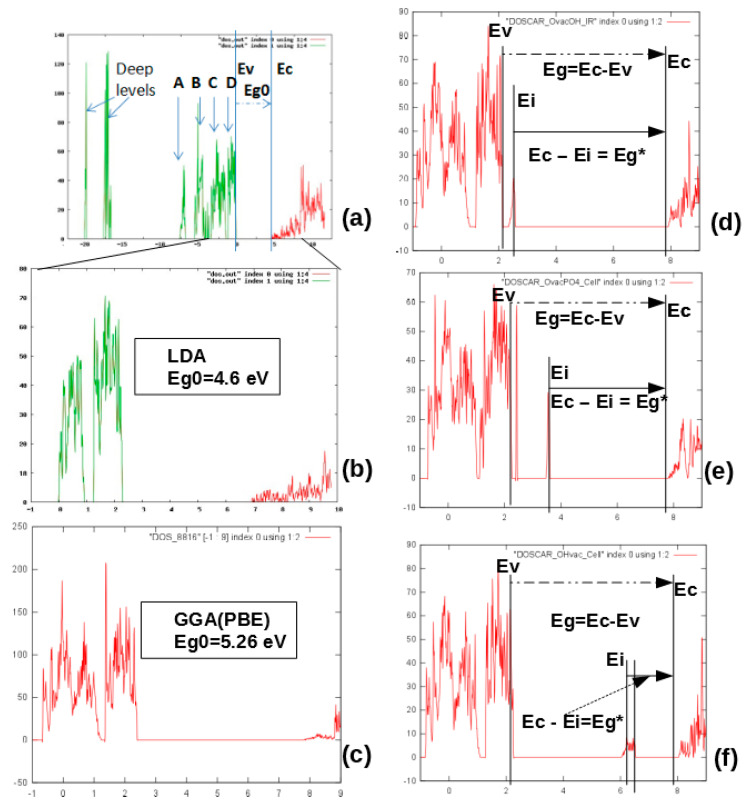
Density of electronic states (DOS) for HAp unit cell: (**a**) initial pure HAp with 44 atoms in hexagonal P6_3_ unit cell lattice, including deep levels and A–D peaks in valence band computed in LDA by AIMPRO; (**b**) the same with main energies around Eg; (**c**) the same computed in GGA (PBE) by VASP; (**d**) DOS in case of O vacancy in OH group; (**e**) DOS in case of O vacancy in PO_4_ group; (**f**) DOS in case of full OH vacancy. (Data presented here obtained after calculations using AIMPRO (LDA) [32] and VASP (GGA) [38], similar to our works [22,28]).

**Figure 12 materials-13-04575-f012:**
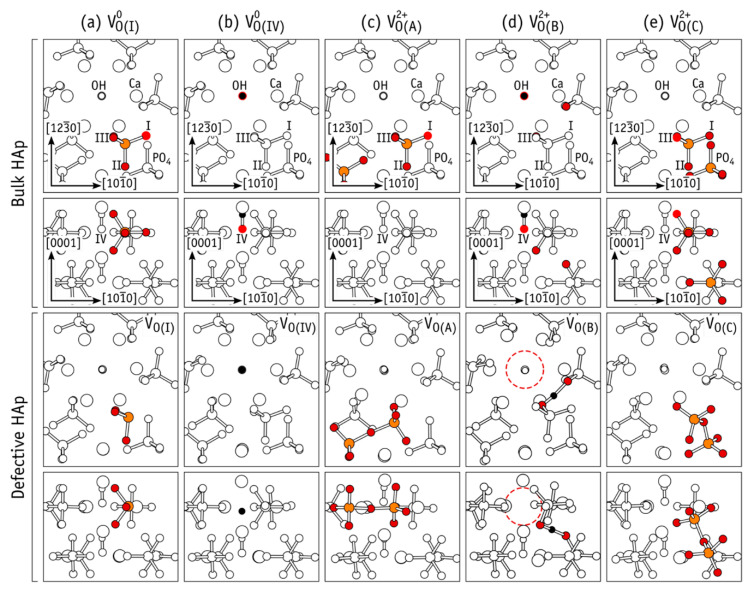
The structure formation of different types oxygen vacancies (V_O(I)_, V_O(IV)_, V_O(A)_, V_O(B)_, and V_O(C)_) are presented in columns (**a**–**e**). The same structures are shown on each pair of rows: along the (1230) (upper row) and (0001) (lower row) directions of the hexagonal lattice. The colored atoms (orange—P, red—O, and black—H) are the core of the defect. The “Bulk HAp” figures demonstrate the vacancies produced by removing oxygen atom (bright red). The “Defective HAp” figures present the resulting structures after atomic relaxation. Reprinted with permission from [35]. Copyright (2019) American Chemical Society.

**Figure 13 materials-13-04575-f013:**
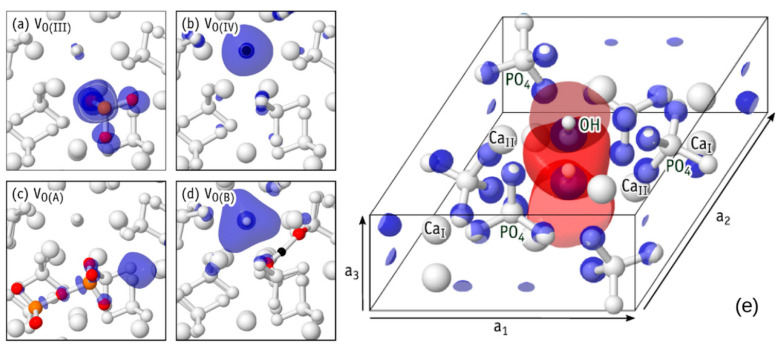
The electron density isosurfaces of the highest occupied Kohn−Sham electron level (for HAp neutral V_O_ vacancy) for: (**a**) V_O(III)_ (for V_O(I)_ and V_O(II)_ are the similar); (**b**) V_O(IV)_, where oxygen O(IV) atom leading to an insulated hydrogen atom in the HAp OH channel; (**c**) V_O(A)_ (extended charged defect); and (**d**) V_O(B)_ (extended charged defect). In all cases, the isosurfaces are shown for the constant level of the electron density n = 0.001 Å^−3^. Reprinted with permission from [35]. Copyright (2019) American Chemical Society. (**e**) The lowest unoccupied Kohn–Sham electron state (bottom of the conduction band) of a HAp at *k* = G. Blue isosurfaces represent y(r) = +0.02 phases of the electron wave function and red ones represent y(r) = 0.02. All atoms are shown in white. Reprinted with permission from [34]. CCC (2018) AIP Publishing.

**Figure 14 materials-13-04575-f014:**
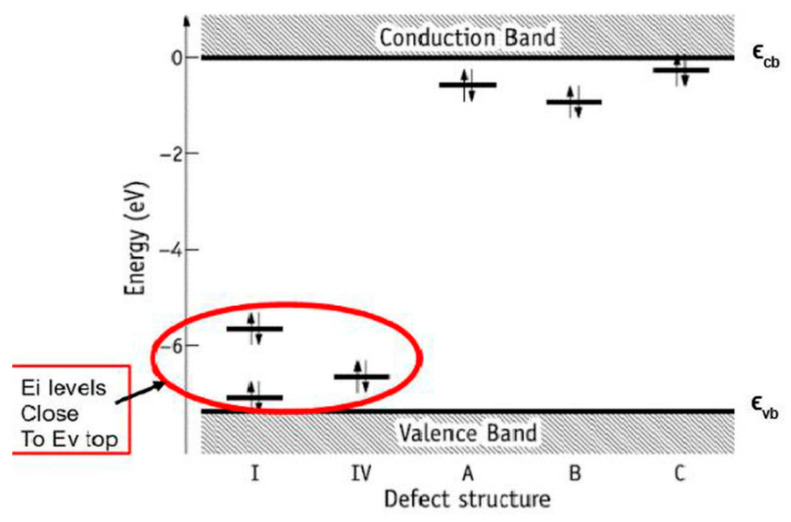
Kohn−Sham energy levels of neutral V_O_ defects in a HAp supercell at the *k* = Γ point. The defect structure I is also representative of structures II and III (see text). The latter have gap states that deviate from those of V_O(I)_ by less than 0.2 eV [35]. Reprinted and modified with permission from [35]. Copyright (2019) American Chemical Society.

**Figure 15 materials-13-04575-f015:**
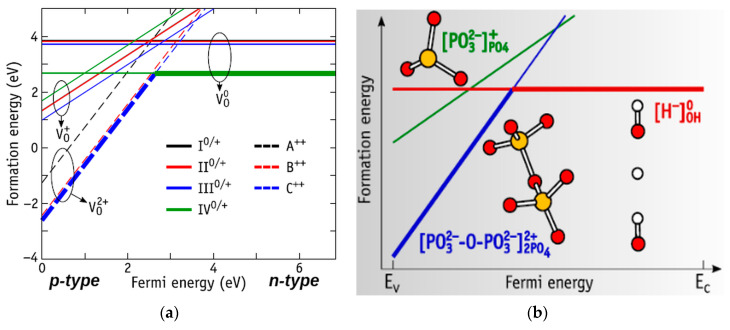
(**a**) Energy diagram of formation of the V_O_ defect in HAp as a function of the Fermi energy. The solid lines represent formation energies of neutral and single positive defects with structures I−IV, while the dashed lines represent double-positive defects with structures A−C. (**b**) Scheme of A^+^/C^++^ defects. Reprinted and modified with permission from [35]. Copyright (2019) American Chemical Society.

**Figure 16 materials-13-04575-f016:**
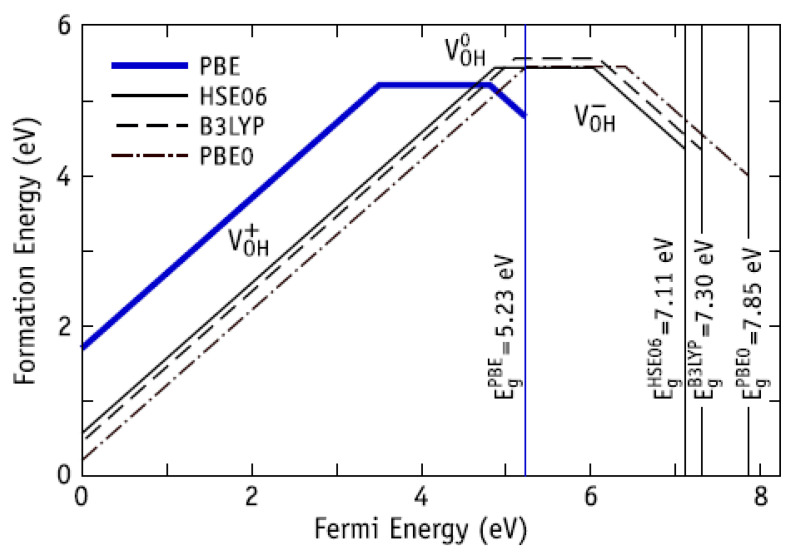
Calculated formation energy of V_OH_ in HAp as a function of the Fermi energy. Positive, neutral, and negative charge states are represented by lines with positive, zero, and negative slope, respectively. The value of Fermi energy varies between the energy value of the top of valence band E_V_ (E_F_ = 0 eV) and the energy value of the bottom of conductive band *E*_C_ (*E*_F_ = ~*E*g). Reprinted with permission from [34]. CCC (2018) AIP Publishing.

**Table 1 materials-13-04575-t001:** Dimensions of the pillars (measured with the above SEM).

Substrate	Cross-section Side, μm	Distance between the Pillars, μm	Number of Pillars on the Single Substrate	Number of Substrates
A	10.4 ± 0.08	13.48 ± 0.12	42,516	4
B	45.46 ± 0.12	12.78 ± 0.16	7248	4
C	9.44 ± 0.08	4.49 ± 0.05	124,439	4

**Table 2 materials-13-04575-t002:** Unit cell parameters *a, b,* and *c* (Å) (from [23]).

Phase	Group	*a,* Å	*b,* Å	*c,* Å
Hexagonal	P6_3_/m	9.417	9.417	6.875
Monoclinic	P2_1_/b	9.480	18.960	6.830

**Table 3 materials-13-04575-t003:** Calculated data for HAp defects by two various approaches (average errors = ±0.05 eV).

Defect type	LDA				GGA (PBE)			
	Eg = Ec − Ev, eV	ΔEg= Eg* − Eg ~Δφ, eV	Ei − Ev*, eV	Ec* − Ei = E_i_^0^, eV	Eg = Ec − Ev, eV	ΔEg = Eg* − Eg ~Δφ, eV	Ei − Ev*, eV	Ec*− Ei = E_i_^0^, eV
HAp in P6_3_/m	4.6	–	–	–	5.26	–	–	–
H vac	4.92	+0.32	0.1 (½ occ.)	4.82	–	–	–	–
O_(OH)_ vac	5.15	+0.55	0.1 (1 occ.)	5.05	5.72	+0.46	0.27	5.45
OH vac	5.49	+0.89	3.11–3.82 peaks: 3.40 3.53 3.66 (½ occ.)	2.38–1.67 peaks: 2.09 1.96 1.83	5.75	+0.49	3.66–4.28 peaks: 3.96 4.11 4.17	1.97–1.35 peaks: 1.78 1.63 1.57
O_(PO4)_ * vac	4.7 ± 0.13	+0.1	1.11 ± 0.3	3.59 ± 0.3	5.34 ± 0.1	+0.08	1.23 ± 0.3	4.11 ± 0.3
H int	5.12	+0.52	Peaks: 1.20 1.45 1.75 (½ occ.)	Peaks: 3.92 3.67 3.37	–	–	–	–
2H _HB Hydrogen filling of unsaturated hydrogen bonds	4.63	+0.03	Peaks: 1.71 0.89 (1 occ.)	Peaks: 2.92 3.74	–	–	–	–

* The averaged values are given here for different positions of O vacancies from the PO_4_ group.

**Table 4 materials-13-04575-t004:** Data for O vacancy from PO_4_ in different position and symmetry (average errors ± 0.05 eV).

Defect type	PBE				B3LYP			
	Eg*= =Ec* − Ev*, eV	ΔEg= =Eg* − Eg~ ~Δφ, eV	Ei − Ev*, eV	Ec* − Ei = = E_ig_^0^, eV	Eg*= =Ec* − Ev*, eV	ΔEg= Eg* − Eg~ ~Δφ, eV	Ei − Ev*, eV	Ec* − Ei = = E_ig_^0^, eV
V_O(I)_	5.0674	−0.1626	1.1496 ~1.15	3.9178	7.0497	−0.2503	1.4291 ~ 1.43	5.6206
V_O(II)_	5.2004	−0.0296	1.3167 ~1.32	3.8837	7.2311	−0.0689	1.6512 ~ 1.65	5.5799
V_O(III)_	5.1393	−0.0907	1.3811 ~1.38	3.7582	7.1333	−0.1667	1.6845 ~ 1.68	5.4488
V_O(IV)_	5.3004	+0.0704	0.4189 ~0.42	4.8815	7.3842	+0.0842	0.7347 ~ 0.73	6.6495
Eg of initial defect-free HAp (P6_3_/m)	5.23	–	–	–	7.3	–	–	–

**Table 5 materials-13-04575-t005:** Treatment types and induced defects.

Induced Defect	Treatments
Vacancy OH	Annealing 500–900 °C [25]
Vacancy O	Gamma-ray irradiation [55]
Vacancy H	Gamma-ray irradiation [55]
Interstitial H	Hydrogenation [56] + MW irradiation
Hydrogen filling of unsaturated hydrogen bonds	Hydrogenation [56]

**Table 6 materials-13-04575-t006:** The plan of the experiment.

Group of Samples	Sequencing						
	PE (Measurement of the Native Specimens)	Annealing	PE (Measurement of the Annealed)	Hydrogenation	MW Irradiation	Gamma-Ray Irradiation	PE (Measurement of the Specimens Processed in 4–6 Sequences)
1	+	+	+	–	–	–	+
2	+	+	+	–	–	+	+
3	+	+	+	+	–	–	+
4	+	+	+	–	+	–	+
5	+	+	+	+	+	–	+

**Table 7 materials-13-04575-t007:** Subgrouping of the samples of the group 1.

Subgroup Number	Calculated φ (Table 3 and [22]), eV	Type of Defect	Measured φ, eV
1	4.6 4.63	HAp without defects 2 additional H atoms that fill unsaturated hydrogen bonds	φ_0_ = 4.51 ± 0.15
2	4.63	2 additional H atoms that fill unsaturated hydrogen bonds	φ_0_ = 4.49 ± 0.09
	4.92	H–vacancy	φ_1_ = 4.84 ± 0.06
3	4.63	2 additional H atoms that fill unsaturated hydrogen bonds	φ_0_ = 4.5 ± 0.1
	5.12 5.15	H–interstitial O–vacancy	φ_1_ = 5.20 ± 0.04
4	4.63	2 additional H atoms that fill unsaturated hydrogen bonds	φ_0_ = 4.47 ± 0.05
	5.49	OH–vacancy	φ_1_ = 5.37 ± 0.05
5	4.92	H–vacancy	φ_0_ = 4.98 ± 0.04
	5.15	O–vacancy	φ_1_ = 5.25 ± 0.04
6	5.12 5.15	H–interstitial O–vacancy	φ_0_ = 5.06 ± 0.04
			φ_1_ = 5.23 ± 0.04

**Table 8 materials-13-04575-t008:** Sub-subgrouping of the samples of the group 1.

Subgroup Number	Sub-Subgroup Number	Calculated φ (Table 3 and [22]), eV	Type of Defect	φ Measured after Annealing, eV
1	1.1	4.63	2 additional H atoms that fill unsaturated hydrogen bonds	φ_0_ = 4.74
		4.92	H–vacancy	φ_1_ = 4.90
	1.2	4.63	2 additional H atoms that fill unsaturated hydrogen bonds	φ_0_ = 4.68
		5.15	O–vacancy	φ_1_ = 5.05
2	2.1	4.63	2 additional H atoms that fill unsaturated hydrogen bonds	φ_0_ = 4.53
		5.15	O–vacancy	φ_1_ = 5.25
	2.2	4.63	2 additional H atoms that fill unsaturated hydrogen bonds	φ_0_ = 4.53
		4.92	H–vacancy	φ_1_ = 5.00
3	3.1	4.63	2 additional H atoms that fill unsaturated hydrogen bonds	φ_0_ = 4.60
		5.49	OH–vacancy	φ_1_ = 5.30
	3.2	4.63	2 additional H atoms that fill unsaturated hydrogen bonds	φ_0_ = 4.60
		5.15	O–vacancy	φ_1_ = 5.20
4	4.1	4.63	2 additional H atoms that fill unsaturated hydrogen bonds	φ_0_ = 4.62
		4.92	H–vacancy	φ_1_ = 5.00
5	5.1	4.63	2 additional H atoms that fill unsaturated hydrogen bonds	φ_0_ = 4.85
		4.92	H–vacancy	φ_1_ = 5.08
6	6.1	4.63	2 additional H atoms that fill unsaturated hydrogen bonds	φ_0_ = 4.70
		5.15	O–vacancy	φ_1_ = 5.15

**Table 9 materials-13-04575-t009:** Subgrouping of the samples of the group 2.

Subgroup Number	Calculated φ (Table 3 and [22]), eV	Type of Defect	Measured φ, eV
2.1	5.49	OH–vacancy	φ_0_ = 5.6
	4.92	H–vacancy	φ_1_ = 4.93
2.2	4.92	H–vacancy	φ_0_ = 4.93
	5.15	O–vacancy	φ_1_ = 5.12

**Table 10 materials-13-04575-t010:** Subgrouping of the samples of the group 3.

Subgroup Number	Calculated φ (Table 3 and [22]), eV	Type of Defect	Measured φ, eV
3.1	5.49	OH–vacancy	φ_0_ = 5.49
	4.92	H–vacancy	φ_1_ = 4.95
	5.15	O–vacancy	φ_2_ = 5.20
3.2	4.63	2 additional H atoms that fill unsaturated hydrogen bonds	φ_0_ = 4.62
	5.49	OH–vacancy	φ_1_ = 5.30

**Table 11 materials-13-04575-t011:** Subgrouping of the samples of the group 4.

Subgroup Number	Calculated φ (Table 3 and [22]), eV	Type of Defect	Measured φ, eV
4.1	4.63	2 additional H atoms that fill unsaturated hydrogen bonds	φ_0_ = 4.50
	4.92	H–vacancy	φ_1_ = 5.00
	5.12 5.15	H–interstitial O–vacancy	φ_2_ = 5.18
4.2	5.12 5.15	H–interstitial O–vacancy	φ_0_ = 5.17
	5.49	OH–vacancy	φ_1_ = 5.33

**Table 12 materials-13-04575-t012:** Subgrouping of the samples of the group 5.

Subgroup Number	Calculated φ (Table 3 and [22]), eV	Type of Defect	Measured φ, eV
5.1	5.49	OH–vacancy	φ_0_ = 5.40
	4.92	H–vacancy	φ_1_ = 5.01
5.2	5.12 5.15	H–interstitial O–vacancy	φ_0_ = 5.20
	4.92	H–vacancy	φ_1_ = 5.04

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
