# Peer review of "Physical Fundamentals of Biomaterials Surface Electrical Functionalization"

_materials, 2020, doi:10.3390/ma13204575_

Round 1
Reviewer 1 Report
Electrical properties of surfaces and surface fictionalization for various purposes are the hot topics of modern material science. (It is nicely reviewed by Authors in the beginning of paper’s Introduction). The authors consider the electrical properties of hydroxyapatite (HA) – well-known material for bioimplants and part of the bones and teeth - as a cargo for cell deposition of proteins, microorganisms, etc. Indeed, this area of research is a kind of a dark matter where a lot can be done. In this sense the paper has novelty and very interesting for the broad audience. It deserves to be published in Materials. Good thing is that the authors tried to connect their theoretical calculations with some original experimental data.
Comments
- Abstract. I suggest to include some concrete results into the Abstract. In present form it looks like a part of Introduction.
- DFT part. Authors are citing either the initial, well-known papers on DFT for calcium phosphates (HA) or their own works. Having nothing against the self-citation, it would be nice to include references on papers of other groups – there are no so many groups in the World making DFT on HA simultaneously with some experiments that support their modeling.
- Fig. 3. (1) I did not quite get the large dispersion of the experimental data. (2) Why some points are bigger than others? (3) Please, change Figure caption. If the word “correlation” is used – please give the correlation parameters. (4) Is it possible to approximate the experimental data with some function?
- Lines 156-157 – Check for upper indexes.
- Line 321. “HAp can be considered as a wide-gap semiconductor”. Please, give any reference where HA-based material with defects really (experimentally) shows semiconducting properties.
Reviewer 2 Report
The authors have prepared a paper with a title suggesting a broad area of interest. As such, the manuscript is ample and very dense in information. However, even though the methodology is well written and many aspects are detailed and considered, the entire paper doesn't come together as a unit. It seems to be made up of 5 sections that share a fundamental theme but rely on separate methods and experiments. Perhaps these should better be published as distinct studies. A serious rewrite could better tie the sections together.
Several other issues occur:
- The introduction (the first one) is long and seems to be written solely in an effort to justify the writing of this article.
- Figure 2 does not read well. I suggest either a color image or thinner lines, connectors, and bullets.
- Figure 8 should be colored, or the bars should have a fill pattern.
- Moderate language editing is needed.
Since the results are sound and the methodology is well established, an improvement or restructuring of the paper could help capitalize on the work put in, but a good form needs to be found.
Respectfully submitted.
Round 2
Reviewer 2 Report
The authors have performed satisfactory changes.
Perhaps there should be some unity in the presentation of graphs and charts (i.e. they should all be provided in full color, if possible).
